# How Water Quality and Quantity Affect Pepper Yield and Postharvest Quality

**Elazar Fallik [1],\*** , **Sharon Alkalai-Tuvia [1]**, **Daniel Chalupowicz [1]**, **Merav Zaaroor-Presman [1]**, **Rivka Offenbach [2]**, **Shabtai Cohen [2] and Effi Tripler [2]**

[1] Agricultural Research Organization, The Volcani Center, Department of Postharvest Science of Fresh Produce, Rishon Leziyyon 7505101, Israel; sharon@volcani.agri.gov.il (S.A.-T.); chalu@volcani.agri.gov.il (D.C.); merav.zaaroor@mail.huji.ac.il (M.Z.-P.)

[2] Central and Northern Arava Research and Development, Arava Sapir 8682500, Israel; Rivka@arava.co.il (R.O.); sab@inter.net.il (S.C.); effi@arava.co.il (E.T.)

\* Correspondence: efallik@volcani.agri.gov.il; Tel.: +972-3-9683665

**Abstract:** There are gaps in our knowledge of the effects of irrigation water quality and amount on yield and postharvest quality of pepper fruit (*Capsicum annuum* L.). We studied the effects of water quality and quantity treatments on pepper fruits during subsequent simulated storage and shelf-life. Total yield decreased with increasing water salinity, but export-quality yield was not significantly different in fruits irrigated with water of either 1.6 or 2.8 dS/m, but there was a 30–35% reduction in export-quality yield following use of water at 4.5 dS/m. Water quantity hardly affected either total or export-quality yield. Water quality but not quantity significantly affected fruit weight loss after 14 days at 7 °C plus three days at 20 °C; irrigation with water at 2.8 dS/m gave the least weight loss. Fruits were significantly firmer after irrigation with good-quality water than with salty water. The saltier the water, the higher was the sugar content. Vitamin C content was not affected by water quality or quantity, but water quality significantly affected antioxidant (AOX) content. The highest AOX activity was found with commercial quality water, the lowest with salty water. Pepper yield benefited by irrigation with fresh water (1.6 dS/m) and was not affected by water quantity, but post-storage fruit quality was maintained better after use of moderately-saline water (2.8 dS/m). Thus, irrigation water with salinity not exceeding 2.8 dS/m will not impair postharvest quality, although the yield will be reduced at this salinity level.

**Keywords:** prolonged storage; salinity; shelf-life

## 1. Introduction

The amount of agricultural land destroyed by salt accumulation each year, worldwide, is estimated to be 10 million ha [1]. Furthermore, this destruction rate could be accelerated by: climate change; excessive use of groundwater; increasing use of low-quality water in irrigation; and the massive introduction of irrigation associated with intensive farming. On the other hand, it has been confirmed in many regions that the tendency to increase the efficiency of irrigation water use and to irrigate with low-quality water, because of water scarcity, can lead to accumulation of salts in the soil. It is estimated that by 2050, 50% of the world's arable land will be affected by salinity [2].

During the last decade, salinity and drought were two of the major abiotic stresses in the Arava Valley in the southern part of Israel. This region is predominantly arid and is affected by salinity because of very low rainfall (<30 mm year$^{-1}$), high evapotranspiration (3000 mm year$^{-1}$), and groundwater that is mostly saline, with an electrical conductivity (EC) about 2.8 dS/m. Moreover, the amount of water available for irrigation is declining every year; salinity is gradually increasing and there

are underground water wells with more than 4 dS/m. Consequently, plant growth and yield can be negatively affected [3]. Azuma et al. [4] reported that the detrimental impact of salinity mainly affects fruits rather than leaves and stems. Thus, high salinity and water scarcity in agricultural soils present the most serious challenges faced by horticultural crops in southern Israel.

The major crop in the Arava Valley during the winter is sweet bell pepper (*Capsicum annuum* L.); about 60% of the sweet bell pepper that is designated for export from Israel is grown in this region during the fall and winter; the growth area is estimated at 2000 ha, with an average yield of about 80–120 ton ha$^{-1}$. Pepper plants are sensitive to drought stress and moderately sensitive to salt stress [5,6]. Nevertheless, very little is known about the influence of water quantity and quality on pepper fruit quality after harvest and prolonged storage. Therefore, the objective of the present study was to evaluate, for two consecutive years, the effects of water quantity (i.e., irrigation water), and quality (i.e., salinity) on pepper yield and fruit quality after prolonged storage and shelf-life simulation.

## 2. Materials and Methods

### 2.1. Plant Materials and Physical Design

The study was performed at Yair experimental station (30°46′45.3″ N; 35°14′31.1″ E) in a 900 m$^2$ greenhouse, situated in Israel's Central Arava Valley, 130 m below mean sea level. The experiment took place during the growing season 2015/2016, in which sweet red bell-pepper (*Capsicum annuum* L., cv. Cannon) was evaluated for yield, fruit quality and postharvest indicators. The local soil texture is loamy sand, having sand, silt and clay percentages of 83, 8 and 9%, respectively [7]. Two row crops of pepper seedlings were planted on 5 August 2015, in each bed and spaced 0.4 × 0.4 m. The distance between each bed was 1.6 m, which yielded a planting density of 31,250 plants·ha$^{-1}$. The experiments were equipped with a pressure-compensated drip irrigation system (Netafim Ltd., Hatzerim, Israel), consisting of one lateral for each crop row having an outer diameter of 0.017 m. The integrated drippers were spaced 0.2 m and their discharge was 1.6 L h$^{-1}$.

Prior to the planting, the greenhouse was enclosed with 25 mesh insect net, with an additional net-shading on the roof which reduced the radiation by 30%. The net shade was removed 6 weeks after the planting, followed by enclosing the greenhouse with translucent plastic (0.12 mm thick, IR—Ginegar Plastic Ltd., Kibbutz Ginegar, Israel), 1 month later. A Spanish trellising method was applied and common cultivation (leaf pruning, side shoots removal, vine-training and canopy-height adjustment) and plant protection practices were used throughout the growing season [8]. Temperature measurements records were downloaded from an adjacent Israeli Meteorological Services (IMS) meteorological weather station.

### 2.2. Irrigation and Yield

The experimental design was randomized blocks (n = 4), with 20 plants in each replicate. Three irrigation water salinities (EC 1, 2.8 and 4 dS·m$^{-1}$) and 3 water application levels were applied for each water quality (Table 1). Irrigation application levels were determined based on the long-term (2002–2014) averages of potential evapotranspiration rates of bell peppers in the Arava region. Electrical conductivity of 1 dS·m$^{-1}$ was applied by blending local saline water (EC = 2.8 dS·m$^{-1}$ with desalinated water, while the highest salinity level (EC of 4.5 dS·m$^{-1}$) was achieved by an equivalent addition of sodium chloride and calcium chloride salts to the local saline water.

**Table 1.** Irrigation water salinities and their specific application levels, since Day After Planting (DAP). Fertilizer solution in irrigation water contained N as total nitrogen, P as $P_2O_5$ and K as $K_2O$.

| | | Electrical Conductivity of the Irrigation Water (dS·m$^{-1}$) | | | | | | | | | | |
| --- | --- | --- | --- | --- | --- | --- | --- | --- | --- | --- | --- | --- |
| | | **1** | | | **2.8** | | | **4** | | | | |
| | | Water Application Levels (% from $ET_p$) | | | | | | | | | Fertilizer Application | |
| **DAP** | $ET_p$ | **70** | **100** | **150** | **100** | **150** | **200** | **100** | **200** | **300** | **N-P-K** | **N** |
| | (mm·d$^{-1}$) | Daily Irrigation Water Depths (mm·d$^{-1}$) | | | | | | | | | (%) | (mg·L$^{-1}$) |
| 0–35 | 1.3 | 0.91 | 1.3 | 1.95 | 1.3 | 1.95 | 2.6 | 1.3 | 2.6 | 3.9 | 6-6-6 | 50 |
| 36–51 | 3.3 | 2.31 | 3.3 | 4.95 | 3.3 | 4.95 | 6.6 | 3.3 | 6.6 | 9.9 | 6-6-6 | 50 |
| 52–62 | 2.7 | 1.89 | 2.7 | 4.05 | 2.7 | 4.05 | 5.4 | 2.7 | 5.4 | 8.1 | 7-3-7 | 120 |
| 63–94 | 2.5 | 1.75 | 2.5 | 3.75 | 2.5 | 3.75 | 5 | 2.5 | 5 | 7.5 | 7-3-7 | 150 |
| 95–104 | 1.7 | 1.19 | 1.7 | 2.55 | 1.7 | 2.55 | 3.4 | 1.7 | 3.4 | 5.1 | 7-3-7 | 100 |
| 105–114 | 1.2 | 0.84 | 1.2 | 1.8 | 1.2 | 1.8 | 2.4 | 1.2 | 2.4 | 3.6 | 7-3-7 | 100 |
| 115–124 | 1.2 | 0.84 | 1.2 | 1.8 | 1.2 | 1.8 | 2.4 | 1.2 | 2.4 | 3.6 | 7-3-7 | 100 |
| 125–134 | 0.8 | 0.56 | 0.8 | 1.2 | 0.8 | 1.2 | 1.6 | 0.8 | 1.6 | 2.4 | 4-2-6 | 100 |
| 135–144 | 0.8 | 0.56 | 0.8 | 1.2 | 0.8 | 1.2 | 1.6 | 0.8 | 1.6 | 2.4 | 4-2-6 | 100 |
| 145–154 | 1.1 | 0.77 | 1.1 | 1.65 | 1.1 | 1.65 | 2.2 | 1.1 | 2.2 | 3.3 | 4-2-6 | 100 |
| 155–164 | 1.3 | 0.91 | 1.3 | 1.95 | 1.3 | 1.95 | 2.6 | 1.3 | 2.6 | 3.9 | 4-2-6 | 100 |
| 165–194 | 2 | 1.4 | 2 | 3 | 2 | 3 | 4 | 2 | 4 | 6 | 4-2-6 | 100 |
| 195–224 | 3 | 2.1 | 3 | 4.5 | 3 | 4.5 | 6 | 3 | 6 | 9 | 4-2-6 | 100 |
| 225–243 | 4 | 2.8 | 4 | 6 | 4 | 6 | 8 | 4 | 8 | 12 | 4-2-6 | 100 |
| 244–272 | 5 | 3.5 | 5 | 7.5 | 5 | 7.5 | 10 | 5 | 10 | 15 | 4-2-6 | 100 |

Yield data included the cumulative weight of fruits (total yield) and that of defect-free fruits (export-quality yield), from December through mid-March from all four repetitions per treatment. Results are expressed in ton ha$^{-1}$.

Petioles were sampled from newly fully-expanded leaves located at the 4th petiole from the apex. Approximately 20–25 petioles were collected at random from each replicate. The samples were taken between 8:00–10:00 am to minimize differences in cell turgidity of plants. The leaflets were stripped, and the petioles placed in a zip-lock bag. One mL of freshly pressed sap was diluted with 50 mL of distilled water. The solution was analyzed for chloride concentration by means of a standard chloridometer instrument.

*2.3. Postharvest Fruit Quality Parameters*

The postharvest quality was determined once monthly, at the end of December, at the beginning of February, and mid-March of each year; there were three harvests per year. Each harvest was collected in four corrugated cartons, each containing 5 kg of export-quality pepper fruits. The fruits were of uniform size of 180–200 g, at 85–90% maturity, with attached calyx and free of defects. Immediately after harvest, fruits were rinsed and brushed in hot water as described by Fallik et al. [9]. Fruit-quality parameters were evaluated immediately after each harvest and at the end of 14 days of storage at 7 °C and relative humidity (RH) of ~95%, followed by 3 days at 20 °C. Weight loss was expressed as percentage loss from the initial weight of 10 fruits. Fruit flexibility was measured by placing the fruit between two horizontal flat plates, the upper of which was loaded with a 2-kg weight, as described by Fallik et al. [9]. A dial fixed to a graduated plate recorded the deformation of the fruit in millimeters. Full deformation was measured 15 s after placing the load on the fruit, the weight was removed, and the residual deformation was measured after a further 15 s. The residual deformation directly indicated fruit elasticity: a fruit with 0–1.5 mm deformation was designated as very firm; with 1.6–3.0 mm deformation as firm; with 3.1–4.5 mm deformation as soft; and with more than 4.6 mm deformation as very soft. Total soluble solids (TSS) were measured in the five fruits that had been tested for firmness, by squeezing juice out of the fruits and recording the readings on an Atago digital refractometer

(Atago, Tokyo, Japan). A fruit was considered decayed if fungal mycelia appeared on the peel or calyx, and decay was expressed as the percentage of decayed fruits in the carton.

The vitamin C content of the bell pepper fruits was determined with the HI3850 Ascorbic Acid Test Kit (Hanna Instruments, Smithfield, RI, USA), which expresses measured quantities as milligrams per 100 g. In accordance with the test kit instructions, 2 g of fresh bell pepper fruit was homogenized with 10 mL of deionized water in a 50-mL vial at high speed for 1 min. The homogenate was passed through filter paper and kept on ice pending mixing of a 1-mL aliquot of homogenate with 49 mL deionized water in a beaker. Then 1 mL of HI3850A-0 reagent and four drops of starch as an indicator were added, and HI3850C-0 reagent was added as 10-mL drops, which were counted until a persistent blue color was developed when the beaker was swirled.

Antioxidant activity (AOX) was measured by using the discoloration method [10] based on 2,2′-azinobis (3-ethylbenzothiazoline-6-sulfonate) ($ABTS^+$) (Sigma-Aldrich, Rehovot, Israel) with slight modification. In the present study, only hydrophilic fractions were isolated from 100 mg of freeze-dried powder by stepwise extraction with acetate buffer, acetone, and hexane, and repeated partitioning of water-soluble and -insoluble portions. Antioxidant activity was evaluated by discoloration of the $ABTS^+$ radical cation. The radical was generated in acetate buffer medium at pH 4.3 to facilitate the activities of the hydrophilic antioxidants. The final reaction mixture contained 150 μmol of $ABTS^+$ and 75 μmol of potassium persulfate ($K_2S_2O_8$) in 249 mL of acetate buffer at pH 4.3. Incubation of the reaction mixture at 45 °C for 1 h was sufficient to generate $ABTS^+$. The resulting stock solution of $ABTS^+$ can be stored for up to 3 days at 4 °C without significant loss of properties. The discoloration test was performed in a 96-well microplate by adding 3 μL of test sample to 300 μL of $ABTS^+$ and comparing the optical density at 734 nm after 15 min of incubation at room temperature, with that of a blank sample. Final results were calculated by comparing the absorbance of the samples with that of the standard (±)-6-hydroxy-2,5,7,8-tetramethylchromane-2-carboxylic acid (Trolox) (Sigma-Aldrich). The antioxidant activity in the samples was determined as Trolox equivalents (TE), according to the formula

$$TE = (A_{sample} - A_{blank})/(A_{standard} - A_{blank}) \times C_{standard}$$

where A is the absorbance at 734 nm and C is the concentration of Trolox (mmol).

The TE antioxidant capacity (TEAC) per unit weight of plant tissue was calculated as follows:

$$TEAC \ (mmol \ TE/mg) = (TE \times V)/(1000 \times M)$$

in which V is the final extract volume and M is the amount of tissue extracted.

The contents of vitamin C and antioxidant activity were measured in 10 fruits taken from each treatment, at each of the three harvests each year.

### 2.4. Statistical Analysis

The data shown here are the means of two consecutive experiments with three harvests each year; the results were similar. The results were subjected to two-way analysis of variance (ANOVA) with JMP 11 version (SAS, Cary, NC, USA). The means were separated by using the Least Significance Difference (LSD) test at *p* < 5%. Pairwise correlation analysis was carried out to determine the significance level of the correlation between the parameters of interest.

## 3. Results

### 3.1. Yield and Chloride in Petiole

The better the water quality, the higher was the total cumulative yield during the growing season (Figure 1); the total yield decreased as the water salinity increased. The average total yield with water of EC 1.6 dS m$^{-1}$ was about 128-ton ha$^{-1}$; in water quality of 2.8 and 4.5 dS m$^{-1}$ the average total yield was 115- and 99-ton ha$^{-1}$, respectively. The export-quality yields were not significantly different

at both 1.6 and 2.8 dS m$^{-1}$, with an average yield of 70- and 65-ton ha$^{-1}$, respectively. Reductions of 35 and 30% in export-quality yield were observed when plants were irrigated with water at 4.5 dS m$^{-1}$, compared with those obtained at 1.6 and 2.8 dS m$^{-1}$, respectively (Figure 1). Water quantity hardly affected either total or export-quality yield, although there was a slight increase in total yield with irrigation at 1.6–1.5 dS m$^{-1}$ and a slight decrease in total yield at 4.5–2.0 or 4.5–3.0 dS m$^{-1}$.

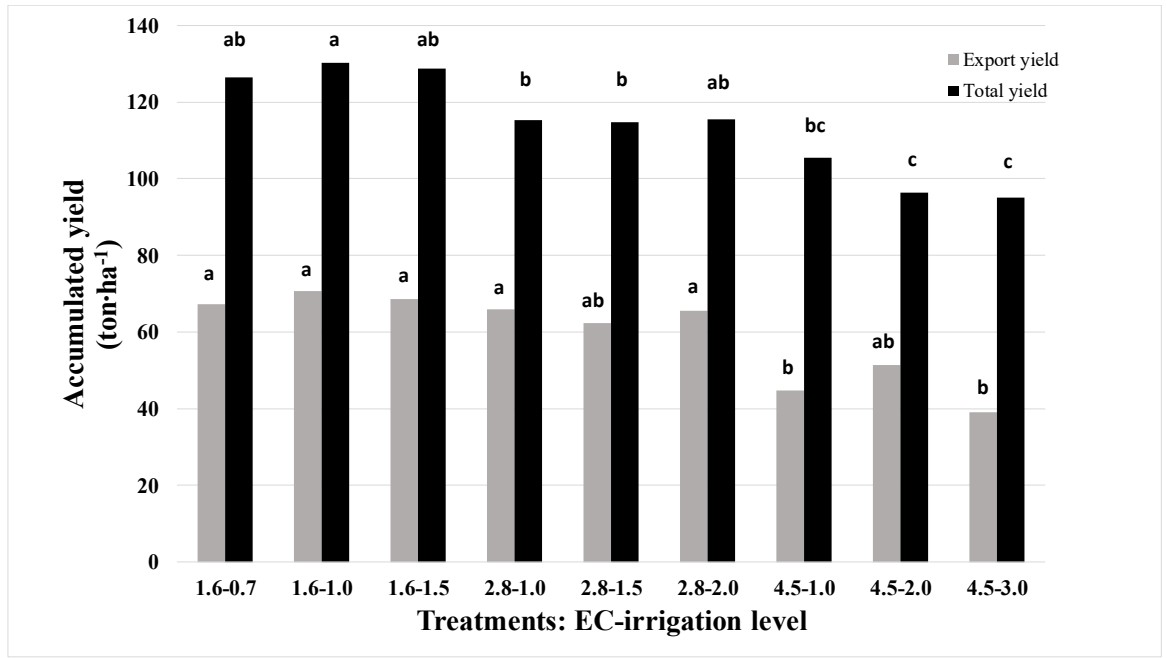

**Figure 1.** The influence of water quality (electrical conductivity (EC) of the irrigation—1.6, 2.8 and 4.5 dS m$^{-1}$ EC) and water quantity on the cumulative total and export-quality yields of pepper between December and mid-March. Means of columns with the same letter are not significantly different (LSD; $p < 0.05$).

High chloride concentrations were reordered in petioles of peppers treated with low irrigation levels (0.7 and 1). Numerically, at salinities of 1, 2.8 and 4.5 dS·m$^{-1}$, the chloride concentrations were 150, 159 and 197.5 mg·L$^{-1}$, respectively. A further increase in irrigation level reduced chloride content in the petioles. However, no differences were observed between the two high water application levels in each salinity treatment (Figure 2).

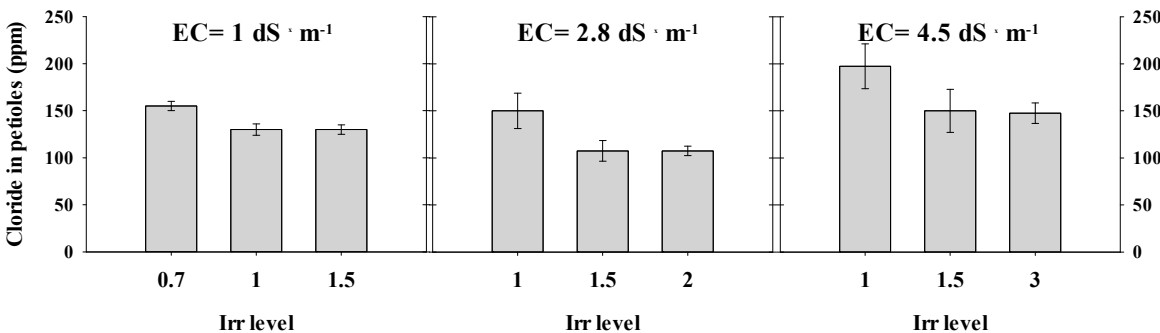

**Figure 2.** Chloride concentration in the petioles of pepper, treated with combinations of various salinities and irrigation (Irr) levels. Measurements were conducted in December 2015. Error bars indicate standard deviation (n = 4).

## 3.2. Fruit Quality

After 14 days at 7 °C and an additional three days at 20 °C, no significant differences between the treatments were observed in percentage loss of fruit weight. However, water quality, but not water quantity, affected fruit weight loss significantly (Table 2, F = 0.04). The better the water quality, the higher the weight loss (an average of 4.03%), while fruit harvested from plants irrigated at EC 2.8 dS m$^{-1}$ had the lowest weight loss (an average of 3.55%) (Table 2). Fruit were significantly firmer (2.6 mm deformation) when irrigated with good quality water (1.6 dS m$^{-1}$ EC), while irrigation with very salty water (EC 4.5 dS m$^{-1}$) gave soft fruits (3.09 mm deformation) (Table 2, F = 0.0058). The TSS was significantly affected by the water quality (Table 1, F = 0.0003); the saltier the water, the higher was the sugar content. The highest TSS content was found in the treatment of 4.5 EC − 3.0 (water salinity − amount of water. See Table 2) (8.72%), while the lowest content was found at 1.6 EC − 1.0 (water salinity − amount of water. See Table 2) (7.53%). No significant differences between the treatments were observed in percentage of decay development, although the highest decay was found in fruit irrigated with 1.6 dS m$^{-1}$ EC (an average of 10.3% decayed fruit) and the lowest decay was found in fruit irrigated with 4.5 dS m$^{-1}$ EC (an average of 7.1%). Water quantities did not affect all fruit quality parameters. No interaction between water quality and quantity was found in relation to external and internal fruit quality shown in Table 2.

**Table 2.** The influence of water quality and irrigation water amount on pepper fruit quality after 14 days at 7 °C plus three days at 20 °C. Means of six harvests during two years.

| Treatment | Water Quality | Amount of Water [z] | Weight Loss (%) [y] | Flexibility (mm) [x] | TSS (%) [w] | Decay (%) |
|---|---|---|---|---|---|---|
| 1 | 1.6 | 0.7 | 4.13 a [v] | 2.70 a | 7.58 b | 14.5 a |
| 2 | 1.6 | 1.0 | 4.05 a | 2.58 a | 7.53 b | 9.3 a |
| 3 | 1.6 | 1.5 | 3.90 a | 2.52 a | 7.55 b | 7.2 a |
| 4 | 2.8 | 1.0 | 3.53 a | 2.32 a | 7.83 ab | 7.5 a |
| 5 | 2.8 | 1.5 | 3.53 a | 2.17 a | 8.13 ab | 6.0 a |
| 6 | 2.8 | 2.0 | 3.58 a | 2.43 a | 8.12 ab | 8.5 a |
| 7 | 4.5 | 1.0 | 3.87 a | 3.12 a | 8.05 ab | 7.0 a |
| 8 | 4.5 | 2.0 | 3.77 a | 3.02 a | 8.37 ab | 7.2 a |
| 9 | 4.5 | 3.0 | 3.62 a | 3.13 a | 8.72 a | 7.2 a |
| **LSD** | | | **0.31** | **0.40** | **0.32** | **5.61** |
| **Mean of water quality** | | | | | | |
| | 1.6 | | 4.03 a | 2.60 ab | 7.56 b | 10.33 a |
| | 2.8 | | 3.55 b | 2.31 b | 8.03 a | 7.33 a |
| | 4.5 | | 3.75 ab | 3.09 a | 8.38 a | 7.11 a |
| **LSD** | | | **0.18** | **0.23** | **0.19** | **3.24** |
| **Mean of amount of water** | | | | | | |
| | | Low | 3.84 a | 2.71 a | 7.82 a | 9.67 a |
| | | Moderate | 3.78 a | 2.59 a | 8.01 a | 7.50 a |
| | | High | 3.70 a | 2.69 a | 8.12 a | 7.61 a |
| **LSD** | | | **0.18** | **0.23** | **0.19** | **3.24** |
| **Analysis of Variance (F-Value)** | | | | | | |
| WQ [u] | | | 0.04 * | 0.0058 *** | 0.0003 *** | 0.54 NS |
| AOW [t] | | | 0.8 NS | 0.86 NS | 0.31 NS | 0.75 NS |
| WA × AOW | | | 0.97 NS | 0.87 NS | 0.61 NS | 0.83 NS |

[z] From evapo-transpiration; [y] Percentage loss from initial weight; [x] Deformation as measured in millimeters; [w] Percentage of total soluble solids (Brix°); [v] Values within each column followed by same letter(s) are not significantly different according to least significance difference test * ($p \leq 0.05$). * $p \leq 0.05$; ** $p \leq 0.01$; *** $p \leq 0.001$; **** $p \leq 0.0001$; NS = non-significant at $p \leq 0.05$; [u] Water quality (WQ); [t] Amount of water (AOW).

Vitamin C content was not affected by water quality or quantity, although fruits harvested from plants irrigated with water at EC 2.8 dS m$^{-1}$ had the highest average vitamin C content (130 mg/100 g FW) compared with the other two water qualities (Table 3). Water quality significantly

affected AOX content in the fruit after 14 days of storage and marketing simulation. The average AOX activity in fruits harvested from plants irrigated at EC of 2.8 dS m$^{-1}$ was 4.6 μM TE/g FW compared with 4.1 and 4.0 μM TE/g FW in fruits irrigated at EC of 1.6 or 4.5 dS m$^{-1}$, respectively. The highest AOX activity was found in the 2.8 dS m$^{-1}$ EC-1.5 treatment (4.8 μM TE/g FW), while the lowest activity was found in the 4.5 dS m$^{-1}$ EC-3.0 treatment (3.9 μM TE/g FW). An interaction was found in AOX activity between the water quality and quantity (F = 0.02) (Table 3).

**Table 3.** Influence of water quality and irrigation water amount on fruit nutritional contents after 14 days at 7 °C plus three days at 20 °C. Means of six harvests over two years.

| Treatment | Water Quality | Amount of Water | Vitamin C (mg/100 g FW) | AOX TEAC (μM TE/g FW) |
|---|---|---|---|---|
| 1 | 1.6 | 0.7 | 121 a [z] | 4.1 cd |
| 2 | 1.6 | 1.0 | 124 a | 4.1 cd |
| 3 | 1.6 | 1.5 | 123 a | 4.3 bcd |
| 4 | 2.8 | 1.0 | 124 a | 4.4 abc |
| 5 | 2.8 | 1.5 | 133 a | 4.8 a |
| 6 | 2.8 | 2.0 | 133 a | 4.6 ab |
| 7 | 4.5 | 1.0 | 126 a | 4.2 bcd |
| 8 | 4.5 | 2.0 | 119 a | 4.0 cd |
| 9 | 4.5 | 3.0 | 118 a | 3.9 d |
| **LSD** | | | **10.6** | **0.13** |
| **Mean of water quality** | | | | |
| | 1.6 | | 123 a | 4.1 b |
| | 2.8 | | 130 a | 4.6 a |
| | 4.5 | | 121 a | 4.0 b |
| **LSD** | | | **6.13** | **0.08** |
| **Mean of water amount** | | | | |
| | Low | | 123 a | 4.2 a |
| | Middle | | 125 a | 4.3 a |
| | High | | 125 a | 4.2 a |
| **LSD** | | | **6.13** | **0.08** |
| **Analysis of Variance (F-Value)** | | | | |
| WQ [y] | | | 0.33 NS | <0.0001 **** |
| AOW [x] | | | 0.96 NS | 0.76 NS |
| WA × AOW | | | 0.81 NS | 0.02 * |

[z] Values within each column followed by same letter(s) are not significantly different according to least significance difference test ($p \leq 0.05$). * $p \leq 0.05$; ** $p \leq 0.01$; *** $p \leq 0.001$; **** $p \leq 0.0001$; NS = non-significant at $p \leq 0.05$. [y] Water quality (WQ). [x] Amount of water (AOW).

In pepper fruit, the correlation coefficient indicated a significantly higher and positive relationship between weight loss and decay development at $p = 0.01$. Weight loss had a significantly high and negative relationship with vitamin C at $p = 0.0001$. Likewise, a negative and significantly higher relationship was also noted between elasticity and decay incidence at $p = 0.01$ (Table 4).

**Table 4.** Correlation coefficients of weight loss (WL), elasticity (Firm), sugar content (TSS), decay, vitamin C (VC) and antioxidant activity (AOX) in red pepper after 14 days at 7 °C plus three days at 20 °C.

| | WL | Firm | TSS | Decay | VC |
|---|---|---|---|---|---|
| Firm | −0.065 | | | | |
| TSS | −0.223 | 0.259 | | | |
| Decay | 0.342 ** | −0.310 ** | 0.193 | | |
| VC | −0.471 **** | 0.099 | −0.014 | −0.202 | |
| AOX | −0.101 | −0.195 | 0.076 | −0.155 | 0.059 |

*, **, ***, and **** = significant at $p$ = 0.05, 0.01, 0.001 and 0.0001 levels, respectively.

## 4. Discussion

Salinity and water scarcity present crucial problems for many crop species in Mediterranean countries where water resources are the main limiting factor. In these countries, the limited water quantities available to farmers and increasing water salinity impair plant growth and yield, which depend on water quantity and quality, and may vary according to the plant genotype [11,12]. Very little is known about the effect of water quantity on postharvest fruit quality, but the influence of water salinity on fruit yield and quality is well-documented; most vegetable crops have a salinity threshold at ≤2.5 dS/m [13]. Pepper plants are categorized as sensitive to moderately sensitive to salinity, although Baath et al. [14] concluded that selected chili pepper cultivars can be irrigated with water of salinity ≤3 dS/m, during at least one growing season.

We have found that water quality was more important than water quantity in determining total and export-quality yields: high water quality (1.6 dS m$^{-1}$) increased yield, whereas high salinity (4.5 dS m$^{-1}$) significantly decreased it; in both cases water quantity did not affect pepper yields. The decrease in total yield caused by salinity was mainly due to decreases in fruit fresh weight and not to the number of fruits per plant (data not shown). A high export quality fraction from the total yield was found when the salinity increased from 1 to 2.8 dS·m$^{-1}$. This can be explained by the lower chloride levels measured in EC 2.8 dS·m$^{-1}$. Previous studies found the same trend. Rameshwaran et al. [15] reported that high salinity reduced pepper yield in two growing seasons and Yasour et al. [16] reported that high water salinity reduced pepper plant biomass and fruit yield in the Arava Valley in Israel. The reduction in total yield at high salinity can be attributed to low water content in the fruit because of poor water uptake at high salt concentration, which affects cell expansion in the growing fruit [17]. It is also possible that the decreased yield and poor fruit quality associated with high salinity are caused by poor photosynthesis, which decreases $CO_2$ availability as a result of diffusion limitations [18], and by decreased $CO_2$ conductance in the stomata and mesophyll [19]. Paranychianakis and Chartzoulakis [20] reported that salt accumulation in the root zone caused development of osmotic stress and disrupted cell ion homeostasis, thereby affecting total yield. However, Urrea-Lopez et al. [21] did not find that habanero pepper fruit yield parameters were significantly affected by low photosynthetic activity associated with water salinity, probably because of the fertilizers used in their experiment.

The best fruit quality, as judged by external and internal quality parameters, after prolonged storage and shelf-life simulation, was found at a water of salinity 2.8 dS m$^{-1}$. Navarro et al. [22] reported that moderately saline water was beneficial when peppers were harvested at the red stage; however, no significant differences in several quality parameters were observed between irrigation with water of 1.6 dS m$^{-1}$ and of 2.8 dS m$^{-1}$. It might be that plants irrigated with fresh water (1.6 dS m$^{-1}$) had large canopies, which evaporated more water, thereby increasing canopy humidity, which would increase postharvest decay development because of Botrytis infection (Table 1). At very high salinity (4.5 dS m$^{-1}$), fruit were softer and more flexible, probably because of severe disturbances in membrane permeability, water channel activity, and stomatal conductance [23]. Salinity increased sugar levels in several crops such as melons, grapes and oranges [24–26]; we have found that the saltier the water, the higher the fruit TSS. The increase in the concentration of these sugars could be due, in part, to a loss of water from the fruit, and/or in part to increased hydrolysis of sucrose, which would yield fructose and glucose, in response to the high osmotic potential in the nutrient solution. The increase in glucose and fructose concentrations could also be associated with an active osmotic adjustment [27]. The increase in sugar level in fruits harvested from high-salinity treatments also could be attributed to the increase in starch biosynthesis in developing fruits, which is believed to increase sink strength [28].

Phytonutrients such as vitamin C or AOX capacity are increasingly important aspects of fruit quality because they are associated with benefits to consumer's health [29], and pepper is considered as one of the healthier fruits [30]. Antioxidant synthesis and accumulation in plants is generally stimulated by biotic or abiotic stress such as salinity; they can protect plant organs from serious oxidative damage to lipids, proteins, and nucleic acids [22]. In the present study, the vitamin C content

was not affected by water quality or quantity, but the highest vitamin C concentration was measured in water of EC 2.8 dS m$^{-1}$. However, in fruits harvested from plants irrigated with water of EC 2.8 dS m$^{-1}$, AOX was significantly higher. These results may indicate that moderate salt treatment may significantly improve the nutritional benefits of the fruit, with respect to prevention of free-radical-related diseases, as reported by Navarro et al. [22]. On the other hand, Ehret et al. [31] reported that AOX in tomato fruits responded more strongly to light and temperature than to water salinity.

In conclusion, pepper yield was increased by fresh water of good quality (EC 1.6 dS m$^{-1}$) and not by water quantity, whereas fruit quality after prolonged storage was better maintained in fruits irrigated with moderately saline water, of EC 2.8 dS m$^{-1}$. Therefore, if the water salinity does not exceed 2.8 dS/m, postharvest quality will not be impaired, although the yield will be reduced at this salinity level. However, if water quality continues to deteriorate and becomes saltier, both pepper yield and postharvest quality will be significantly affected.

**Author Contributions:** E.F. was the head of the project; he planned the research, analyzed the results and wrote the manuscript with the rest of the team. D.C., S.A.-T. and M.Z.-P. are research engineers in Elazar Fallik's laboratory; they conducted the experiments, evaluated fruit quality, and analyzed the data in both 2016 and 2017. R.O., S.C., and E.T. were in charge of the planning, and growing practices, yield evaluation and harvest.

**Funding:** This research was funded by the Chief Scientist of Israel's Ministry of Agriculture and Rural Development; project No. 430-0511-14/15/16.

**Conflicts of Interest:** The authors declare no conflict of interest.

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
