# Peer review of "How Water Quality and Quantity Affect Pepper Yield and Postharvest Quality"

_horticulturae, doi:10.3390/horticulturae5010004_

Round 1

Reviewer 1 Report

Dear editor! 

The manuscript aim to evaluate the effect of water quality and quantity on pepper yield and postharvest quality by .Elazar Fallik, Shalon Alkalai-Tuvia, Daniel Chalupowicz, Merav Zaaroor-Presman, Rivka Offenbach, Shabtai Cohen and Effi Tripler. Introduction part is focused on the paper, methodologies are given in detail, and results presentation, interpretation and discussion is satisfactory. 

Section results and discussion needs attention, I suggest that authors evaluate correlations between studied parameters and include them into results and explain briefly in discussion. For example, correlation between vitamin C and AOX is around 0,90, between vitamin C and flexibility is around - 0,65. I am sure that would be of interest for readers.

Best regards

Author Response

An additional explanation was added in lines 176-177 regarding the correlation analysis that was carried out and Table 4 that was added based on reviewer 1 

Reviewer 2 Report

The article studies an important agricultural problem, saline irrigation water, in a production environment and on a production scale. It attempts to link production results to fundamental physiological principles. This approach is certainly to be recommended, however additional information is necessary to correctly situate the results and make them usefull for both farmers and other researchers.

Major remarks

-          It seems necessary to provide a mineral analysis  (at least macronutrients, Na and Cl) of the used water(s), as electrical conductivity of ground water (I assume ground water was used from the text)? in arid regions might be derived  from various ions – possibly not all NaCl, but nutrients as Ca and Mg as well. Their effect on ‘salinity stress’ would be different from NaCl.

-          Figure 1 does not state significant differences (p.e. displayed by letters)? In all results tables and figures, provide standard deviation + sample size

-          Please elaborate on the ‘export quality’ grade: which are types of defects observed rendering fruits unfit for this category and how are they linked to salinity/drought stress?

-          Please provide additional information on nutrient application  during the experiment

-          Please provide additional information on soil type category

-          Please specify ‘decay’: postharvest fungal disease only caused by botrytis? Provide information on timing and type of fungicide treatments.  Might reduced decay be linked to (albeit apparently non-significant) elevated AOX in certain treatments?

Minor remarks

-          EC measurements need to be expressed together with the reference temperature used, or at least reference temperature needs to be mentioned once in ‘material and methods’. Check EC meter settings. For example, “2mS/cm (at 20°C)”

Suggestions (future research)

-          Future research on this topic might benefit from monitoring (salinity-induced) drought stress by means of continuous soil water potential measurement / soil EC measurement.

-          Multiyear trials might provide additional information of sustainability of irrigation practices in regards to (possibly deteriating) soil quality.

Author Response

Major remarks

Remark # 1 - chlorine concentration in the petioles was added in Fig. 2.

Remark # 2 - Letters indicating statistical differences were added in Fig 1.

Remark # 3 - Fertilizer application dosages were added and can be found in Table 1.

Remark # 4 - Soil type was added in Materials and Methods - lines 75-76.

Minor remarks

Remark # 5 - Today every electrical conductivity sensor is equipped with temperature compensation procedure.

Suggestions

a research in this direction has been initiated recently
